# Programmable Density of Laser Additive Manufactured Parts by Considering an Inverse Problem

**DOI:** 10.3390/ma15207090

**Published:** 2022-10-12

**Authors:** Mika León Altmann, Stefan Bosse, Christian Werner, Rainer Fechte-Heinen, Anastasiya Toenjes

**Affiliations:** 1Leibniz-Institute for Materials Engineering-IWT, Badgasteiner Str. 3, 28359 Bremen, Germany; 2Department of Mathematics and Computer Science, University of Bremen, Bibliothekstr. 5, 28359 Bremen, Germany; 3Faculty of Production Engineering, University of Bremen, Bibliothekstr. 1, 28359 Bremen, Germany; 4MAPEX Centre for Materials and Processes, University of Bremen, 28359 Bremen, Germany

**Keywords:** additive manufacturing, LPBF, PBF-LB/M, AlSi10Mg, machine learning, inverse problem, adjustable relative density, demand-oriented process parameters

## Abstract

In this Article, the targeted adjustment of the relative density of laser additive manufactured components made of AlSi10Mg is considered. The interest in demand-oriented process parameters is steadily increasing. Thus, shorter process times and lower unit costs can be achieved with decreasing component densities. Especially when hot isostatic pressing is considered as a post-processing step. In order to be able to generate process parameters automatically, a model hypothesis is learned via artificial neural networks (ANN) for a density range from 70% to almost 100%, based on a synthetic dataset with equally distributed process parameters and a statistical test series with 256 full factorial combined instances. This allows the achievable relative density to be predicted from given process parameters. Based on the best model, a database approach and supervised training of concatenated ANNs are developed to solve the inverse parameter prediction problem for a target density. In this way, it is possible to generate a parameter prediction model for the high-dimensional result space through constraints that are shown with synthetic test data sets. The presented concatenated ANN model is able to reproduce the origin distribution. The relative density of synthetic data can be predicted with an R^2^-value of 0.98. The mean build rate can be increased by 12% with the formulation of a hint during the backward model training. The application of the experimental data shows increased fuzziness related to the big data gaps and a small number of instances. For practical use, this algorithm could be trained on increased data sets and can be expanded by properties such as surface quality, residual stress, or mechanical strength. With knowledge of the necessary (mechanical) properties of the components, the model can be used to generate appropriate process parameters. This way, the processing time and the amount of scrap parts can be reduced.

## 1. Introduction

Due to the possibility of the nearly unconstrained freedom of design and the low amount of wasted material the additive manufacturing process spreads into many industries. Further, the production costs as well as the production time can be decreased using additive manufacturing techniques [1]. One commonly known process is the laser powder bed fusion process (LPBF) also known as powder bed fusion of metal with a laser beam (PBF-LB) or selective laser melting (SLM), a powder-based laser additive manufacturing process [2,3]. Further industrial-relevant processes are electron beam melting (EBM) and laser metal deposition (LMD) [4]. Parts built in these types of processes consist of multiple layers erected one after another [2]. In three steps, the whole process can be described: Pre-process (preparation of the 3D data), manufacturing process (building the part), and post-process (heat treatment and surface finishing) [2]. The cost of additive manufactured parts is mainly driven by the machining time and the material costs. Thus, the needed machine time accounts to more than half of the total costs [5]. This depends on the build rate (BR), further on the time per layer, and the layer thickness [5,6]. State-of-the-art process parameters are chosen to reach the highest possible relative density and mechanical properties [2]. The relative density highly depends on the process parameters as well as the powders quality. During the process, pores arise due to spontaneous interruptions of the melt tracks and not entirely molten powder particles or gassed material. Commonly, the induced energy density (ED) is used to appraise the resulting density without considering physical effects such as the depth of the melt track and the prior mechanisms [7]. The ED is an indication if the material exhibits unmolten areas or even gassed ones. These pores, especially with sharp contours resulting from low EDs, can act as a starting point for cracks and reduce the reachable mechanical properties [8]. As shown by Bertoli et al. [7] for 316 L, the same ED values can be obtained with different process parameter combinations. Further, each process parameter influences the resulting relative density reducing the significance of the ED and can obtain misleading conclusions. Thus, the BR is subordinated to the resulting relative density, which causes too high safety factors for less stressed parts and part areas. Especially, if no high mechanical requirements exist [9]. Parts with densities higher than 99% have comparable mechanical properties to conventional casted or milled parts which are not always needed [2]. In the process, relative densities over 99% can become obsolete if hot isostatic pressing (HIP) is applied afterwards. Hereby the pores resulting from the LPBF process are closed by applying temperature and pressure. At a critical temperature, the external pressure exceeds the yield strength of the printed material and consequently leads to densification. Thus, parts with relative densities of 95% can be densified to 99.8% opening a wide range for improving the BR of the LPBF process [10]. This way, Herzog et al. [10] were able to increase the scan speed by 2/3 with slightly higher residual stresses. Through demand-oriented process parameters, the BR can be increased for less important part areas allowing significantly lower machine times. Demand-oriented process parameters for parts or part areas are current research topics. This way, e.g., more complex geometries can build with homogenous properties (density, surface roughness) due to locally different heat transfer [11]. For practical use the LPBF manufactured parts need specific properties which can be set by the process parameters adequately [12]. One approach for process parameter prediction is presented by Park et al. [13].

High relative densities can be reached by a variety of significantly different process parameters [7]. Bai et al. show that the hatch distance is a key factor when the layer thickness is constant. This way, a variety of laser powers and scan speeds can be chosen to reach the same density values. Small hatch distances further improve the mechanical properties of the parts [14]. Increasing the layer thickness from 40 µm to 50 µm the manufacturing process is much faster with nearly identical mechanical properties [15]. Increasing the layer thickness has no effect on the final phases of 18Ni-300 (MS1 maraging Steel), while changing other process parameters results in austenite volume changes [15]. To reach targeted relative densities, the parameters for the LPBF process have to be chosen. Large experimental designs are needed to discover the resulting densities because each parameter (continuously adjustable within the limits of the LPBF machine specification) as well as the interactions influences the results [2]. Predicting the relative density for known parameters is constrained by the complexity of the interactions of the parameters. Machine learning models can be trained to solve this problem allowing more precise predictions of the resulting relative density [13,16]. However, they also need a great amount of experimental data. This could be reasonable in fact that the models can be adapted for different materials or further mechanical properties with decreasing amounts of data needed. Additionally, machine learning algorithms are able to learn physical connections only from data, enabling them to learn very complex regressions and connections without the need for a priori assumptions [17]. Minbashian et al. [17] showed that artificial neural network (ANN) models obtain accuracies such as the most complex multiple regression models with a high amount of included terms. The inverse problem must be taken into account when generating parameters automatically [13,18]. As a result of the complex interactions, a nearly infinite amount of possible parameter combinations for one specific density value exists [14]. Common machine learning algorithms are not able to learn a problem with a higher dimension of outputs than inputs [18]. One solution could be recommendation algorithms used for choosing from an outsized number of possibilities. Most of them return recommendations by a single domain, e.g., the relative density. Yu et al. presented a support vector machine approach for a multi-domain recommendation algorithm [19]. By adding bound vectors, support vector machines can be improved [20]. By adapted and concatenated machine learning methods, this mathematically underdetermined inverse problem can be solved by taking up the forward model [13,18,21]. This concatenation is based on the structure of generative adversarial networks (GAN), in which a discriminator classifies data sets generated by a generator as real or synthetic. Both models are trained together, so that after the training is completed, artificial data is generated which can no longer be distinguished from real data. In this context, GANs belong to unsupervised learning methods [22]. By concatenating the backward model with the trained forward model, the model can be trained and supervised to a given target value. By modifying the loss function, hints can be used to improve the generated parameters with respect to the BR.

From a materials engineering perspective, the LPBF process can produce a high number of specimens in a relatively short time. The amount of data points is restricted by the subsequent metallographic processes. These processes include grinding, polishing, and microscopic imaging to provide metallurgical micrographs of each specimen. However, from a machine learning perspective, the amount of provided data set is relatively small. Therefore, the used ANNs are relatively compact, which should be able to cope with a smaller amount of data for the training process. This paper is a first serve of applying machine learning models in material sciences and deriving a deeper process understanding within the field of LPBF. Therefore, a statistical test series represented by a full factorial design should create the fundamental data set for this paper. This way, it should be examined how well ANNs perform on small statistical test series in predicting material properties.

In the following, the work is split into four sections. First, the inverse problems with data-driven methods are described. The Materials and Methods section includes specimen manufacturing and analysis, process parameter selection and evaluation, data analyses, and the model-building process used in this work. Afterwards, the results for a theoretic and real data-driven model are presented and discussed. Finally, conclusions are drawn. It is shown that LPBF components made of AlSi10Mg can be manufactured with a defined relative density. For this purpose, the inverse problem is solved.

## 2. Inverse Problems with Data-Driven Methods

Inverse problems can be found in a wide range of science and engineering applications, with the aim to infer an unknown quantity *F* that is not accessible (observable) directly. There is only access to another observable quantity *G* that is related to it via a linear or a non-linear relation *H* [23], such that *G* = *H*(*F*). Solving inverse problems with hidden, complex, or partially unknown physics is often expensive and requires different formulations and simulations [24]. Inverse problems are mainly based on regularization theory and commonly use discretized representations of the unknown model function [23]. Discretization introduces an error ε, such that *G* = *H*(*F*) + ε. Probabilistic Bayesian-inference-based methods are used for solving inverse problems, too, especially if the observable measurement data is noisy, uncertain, or some are missing and with some outliers. A principle discussion of regularization, Bayesian methods, and the benefit of machine learning can be found in [23]. In addition to Bayesian optimization, finding an optimum of a function *F* without any assumptions about the form of *F*, in [25], generative methods and global optimization are proposed to solve inverse problems (such as material design), based on data-driven methods.

Simulation-based approaches, additionally combined with experimental data, using, e.g., physical informed neural networks (PINN), are suitable to solve inverse problems [24,26,27], but the required physical differential equations are currently not accessible for such a complex material process considered in this work, or at least they would oversimplify the problem in a way making this approach unsuitable for process parameter predictions.

Typical data-driven modeling maps a set of observables *x* as input variables on feature variables *y* as the output. The model function *f*(*x*): *x* → *y* derived from measured (or simulated) data contains hidden system parameters of interest, e.g., process parameters, which can be contained in *x* for generalization. The system parameters of interest cannot be measured directly [21], and moreover, the output features are inductive knowledge retrieved by inference, but in methodological science, deductive knowledge is required. For process optimization, e.g., the system parameters given for a specific feature *y* are of interest, defining an inverse problem *f*^−1^(*y*): *y* → *x*. The system and input parameters *x* are typically metric variables, whereas the output feature variables can be metric or categorical variables.

A model function *f*(*x*): *x* → *y* is bijective if it is invertible and there is a (unique) bidirectional mapping of *x* ⇔ *y*, i.e., for each *x* there is a *y*, and for each *y* there is an *x*, e.g., *f*(*x*) = *x* + 1. Linear functions are commonly invertible. In the case of multivariate functions, a set of equations is necessary to solve the inversion analytically. A solver that inverts a function *f* should assess the diversity of possible inverse solutions (if not unique) for a given measurement and should be able to estimate the complete posterior distribution of the system parameter [21].

Complete inversion with unique solutions is a challenge, and even a simple perceptron (artificial neuron) cannot be inverted without additional constraints or auxiliary variables:(1)$$\begin{array}{l} y=f\left(\vec{x}\right)=g\left(\sum_{i}x_{i}\cdot w_{i}+b\right)\\ y=s\left(\vec{x}\right)=\sum_{i}x_{i}\cdot w_{i}\to s^{-1}=\left\{x_{i}=\frac{y-b-\sum_{j\neq i}x_{j}w_{j}}{w_{i}}\right\}\\ y=g\left(x\right)=k\cdot x\to g^{-1}=\frac{y}{k}\\ y=g\left(x\right)=\frac{1}{1+e^{-x}}\to g^{-1}=\ln\left(\frac{y}{1-y}\right) \end{array}$$

Here *w**_i_* are internal model parameters (not to be confused with system parameters) and *g* is the transfer function, e.g., a linear, rectified linear, or the saturating sigmoid-logistics function. Inversion of functions can introduce poles and high non-linearities. The transfer function can be commonly fully or partially invertible, but the inversion of the sum results in a set of equations with an infinite set of solutions. The main reason for this non-solvable inversion problem is the information and dimensionality reduction in a perceptron (ℝ^n^ → ℝ). The strong information reduction is typical for ANN creating an ill-posed problem space. This under-determined problem can be solved by using multiple combined and overlapped perceptrons, but requires still that the dimensions of *x* and *y* are equal, i.e., |*x*| = |*y*|.

With respect to data-driven and experimental methods, the forward path *x* → *y* is easily accessible, but the backward path *y* → *x* is initially hidden. Using invertible functional graph networks, these networks can be trained by the data-driven approach. Invertible neural networks (INN) satisfy three major features: The mapping between input and output is bijective, both forward and backward computations are efficient, and direct computation of the posterior probabilities is possible [13]. To address the information loss in many data-driven problems, a latent and auxiliary output variable *z* is introduced normalizing the input and output dimensions, i.e., |*x*| = |*y*| + |*z*| and *x* ↔ [*y*,*z*] become a bijective mapping. The well-posed forward path can be trained and supervised, instead of the ill-posed inverse problem, see Figure 1b. It is required that the latent variable *z* is independent of *y*, and should follow an easy distribution, e.g., *N*(0,1). The inverse model is basically a conditional and tractable probability *p*(*x*|*y*), but the derivation from the forward model is not tractable (being just an approximation by the latent probability distribution *p*(*z*) and *z* ∈ ℝ^k^ and multi-variate normal distribution *N*). The latent variable distribution is derived in a training process by, e.g., the maximum mean discrepancy (MMD), which is a kernel-based method for comparison of two probability distributions that are only accessible through samples [21]. To train such an *x*/*y*/*z* network, the supervised data loss *L*_y_ and the unsupervised latent space loss *L*_z_ must be considered in the optimization process. If both losses reach zero, *g* = *f*^−1^ returns the true posterior *p*(*x*,*y*) for any *y*. For improved convergence, an additional loss *L*_x_ on the input side can be used and implemented again by MMD [21]. So, INNs are basically auto-encoders whose codes have the same size as the original data.

An invertible network is typically composed of two independent but coupled ANN, one for the forward and one for the backward path. A simple chain of the forward and backward model is similar to the auto-encoder, whereas in [28] an affine coupling layer is used to embed multiple ANN models for the forward and backward paths by using latent variable pairs [*u*_1_, *v*_1_] and [*u*_2_, *v*_2_]. Every single ANN must not be invertible. Both architectures are compared in Figure 1d,e. The training process, such as in the proposed method in this work, is a closed-loop iterative process incorporating the forward and backward model with its specific loss functions as described above.

However, even if we have a suitable invertible architecture, e.g., an invertible neural network, training of deep neural networks requires a big amount of data in terms of diversity, variance, noise, and completeness of the input and output space, not always available for scientific problems such as the one addressed in this work [24]. Additionally, an increase in the data volume can result in a decrease in physical correlation, see Figure 1a. So, even solving the forward path can be challenging, and forward models derived from incomplete and sparse data sets lack the required generalization and are especially important for the inversion. Additional constraints derived from physical laws can support the training and optimization process by limiting the hyper-parameter space of the optimization problem (minimizing the overall |*y*_o_−*y*| error). Physics-informed learning integrates (noisy) data and mathematical models and implements them through neural networks or other kernel-based regression networks [24]. Both, the information loss of ANN and the contradiction of data amount and correlation with physics characterize the problem to be solved in this work, shown in Figure 1. 

A lot of physical models are related to partial differential equations (PDE). The basic concept, as an extension of the INN, is the combination of an ANN outputting intermediate (code) features that are passed to a PDE, which is part of the training loss path, see Figure 1c. Although the data set used in this work poses strong sparseness, parameter gaps, bias, and noise, physics-informed methods are not applied here due to the (unknown) complex and probabilistic material models arising in additive manufacturing processes. Even simulation of the additive manufacturing process is a challenge with respect to computational complexity and matching real-world physics.

## 3. Materials and Methods

### 3.1. LPBF and Metallography

The base of the examination is a statistical experiment design with a full factorial combination of four steps per parameter printed on an SLM 125 HL (SLM Solutions Group, Lübeck, Germany) in common aluminum alloy AlSi10Mg, see Table 1. The maximum laser power used for this work is set at 350 W. Parameter programming and print file preparation were performed in Materialise Magics (Materialise GmbH, Bremen, Germany). The influence of the laser power *P_L_*, the hatch distance *h_S_*, the scan speed *v_S_*, and the layer thickness *D_S_* are investigated. The layer thicknesses are chosen so that two-layer thicknesses can be printed in one job by using the skip layer function, which reduces the amount of build jobs. 

The rest of the process parameters were chosen in even steps constrained by minimum and maximum values representing extreme parameter selections. In this way, it should be possible to achieve both very low and very high relative densities. The EDs studied range from 2.92 J/mm^3^ to 340 J/mm^3^. All relative densities are measured by image analysis of metallurgical micrographs. To speed up the analytical process, three process parameter sets are combined in one specimen. It can be guessed that the three segments in each specimen are mutually influenced through heat conduction. However, it can be observed in Figure 2, that there is no significant influence of the heat conduction when using multiple parameters in one geometry. The selected region of interest (ROI) in the core of each segment for relative density measurement should be a representative value without the influence of thermal conduction or contour parameters. Additionally, the process parameters in each specimen are randomized to avoid spatial dependencies due to heat conduction.

This way, 128 process parameter combinations can be printed in 43 specimens. By using the skip layer function (layer thicknesses of 40 µm and 80 µm, as well as 50 µm and 100 µm, are printed in one run), only two print jobs with 43 specimens each are needed to produce 256 combinations. After printing, the specimens are embedded in a polymer, ground, and polished parallel to the build direction before micrographs are taken. Thus, lower relative densities can only be measured with an increasing error caused by the removal of unsolidified powder particles during grinding.

### 3.2. Data Analysis and Preparation

The most common use for machine learning algorithms is the identification of patterns in large data sets or regression functions where, e.g., no physical laws are known [29]. Conversely this means that a sufficiently large number of data points is needed for the training process. For this reason, it is necessary to extend the aforementioned data set with results from other projects, resulting in more than 400 instances. Differences in the AlSi10Mg powder used as well as the particle distribution are not considered in this work. Afterwards, statistical methods are applied to identify the data sets’ quality which is linked to the achievable model quality [30]. Based on these results, the data set will be prepared for the model training process. Removing missing data and invalid as well as valid outliers should improve the quality of the trained models [16,31]. Missing data points can result from process parameters that do not allow the fabrication of dense specimens that can be analyzed by micrographs. Further invalid outliers are, e.g., process parameter values that cannot be set in the used machine or very low relative densities that cannot be reliably measured with the methods used. Valid outliers will be identified by a quartile analysis [32]. Knowing valid outliers, the range of interesting relative densities can be set. It can be assumed that relative densities below 70% are not technologically useful. 

As common, the net data set will be scaled linearly into the same value interval for all attributes and targets afterwards [33]. The small net number of instances (350) is increased by normal distributed fuzziness for the process parameters. A simple function for the standard deviation with respect to the relative density is established by a quick check of the process stability. This allows the data set to be expanded to 700 instances.

### 3.3. Modeling

For the modeling process, ANNs are prepared using the python framework PyTorch [34]. Due to the dynamic graphs in PyTorch, loss values can be manipulated before they are backpropagated through the model. To check if the forward and especially the backward modeling problem can be solved by concatenated ANNs and manipulating the loss value, artificial data will be used. Afterwards the models can be trained on the real data described before.

As mentioned, the forward modeling problem and the inverse modeling problem, see Figure 3, are addressed in this article. To solve the forward prediction problem, resulting in relative density for known process parameters, common supervised ANNs [35] will be trained on the prepared database. As usual, the input dimension is higher than the output dimension. Predicting the relative density resulting from a given set of LPBF process parameters is fundamental for the inverse modeling process. Thus, the influence of each process parameter needs to be modeled to be able to solve the inverse problem. This is condensed in the following hypothesis.

**Hypothesis** **1.***By the use of machine learning algorithms, the resulting relative density of given process parameters can be predicted even no direct correlation exists*.

Solving the inverse problem, process parameter prediction for a target density should firstly be achieved by creating a database with the best trained forward model as shown in Figure 4 and introduced by Park et al. [13]. Each factor is subdivided into steps regarding the value interval defined in Table 1. With a full factorial combination, a parameter space with up to 100 k combinations will be created. Through the trained forward model, the densities of each combination can be predicted. Detecting the best parameter sets occurs by searching algorithms and the build rate as a constraint.

**Hypothesis** **2.1.**
*With the trained forward model, a database can be created that contains a finite amount of possible solutions for the backward modeling problem. Using a search algorithm and constrained by the build rate, process parameters for specific density values, and layer thicknesses can be identified.*


Secondly, the inverse modeling problem can be solved by concatenating the forward model with an inverse ANN architecture, see Figure 5. This principle is oriented to the functionality of GANs. The inverse ANN generates a set of process parameters. Subsequently, the trained forward model predicts the resulting density. This manipulation allows a comparison of the target density and the resulting one. In this step, the forward model acts as a simple calculation in which operations are written to the dynamic graph of the variable. By following the dynamic graph backwards, the gradients are set and the loss value can be unfolded. This allows the regular not trainable inverse problem to be trained.

**Hypothesis** **2.2.**
*By the use of the dynamic graphs in PyTorch, an inverse architecture can be concatenated with a trained forward model. This way, it is possible to train an ANN with a lower dimension of the inputs than the outputs. Furthermore, the process parameters can be optimized regarding the achieved build rate by adding a hint to the loss function.*


## 4. Results and Discussion

### 4.1. Examination of the LPBF Process

The experimental test series shows the process limits at low as well as high EDs. Overall, 242 out of 256 parameter combinations are possible to build. Those that cannot be built have ED values < 15 J/mm^3^ and track energies < 1 J/mm. Overall densities in the range of 20% to nearly 100% were produced. Three classes of pore formations can be spotted. In Figure 6 are metallurgic micrographs mapped by the laser power and scan speed for two-layer thicknesses and three hatch distances. First, prior unshaped pores can be detected accompanied by low relative densities, colored blue in Figure 6. Second, there are big prior round pores accompanied by medium to high relative densities, colored grey in Figure 6. Third, there are micrographs with small round and occasional unshaped pores reaching the highest mean density, colored orange in Figure 6. By decreasing the volume energy (lower laser power and higher scan speeds), big unshaped pores of the blue class occur. At very high-volume energies (low scan speed with high laser power), big round pores of the grey class result. Decreasing the hatch distance increases the number of orange classified as high-density results at the expense of the blue class, see Figure 6. By increasing the layer thickness from 40 µm to 100 µm, more process parameters with lower laser power cannot be built up. Moreover, the amount of orange classified micrographs decreases significantly, see Figure 6.

The three classes of micrographs depend on the shape of the pores correlating to the pore-building mechanisms. The prior unshaped pores (blue) are caused by the lack of a fusion mechanism. The introduced energy is too low for melting the layers entirely, resulting in poor connections between the layers [36]. Large and prior round pores (grey) result from the keyhole effect. Thus, the energy introduced is high enough to gas/gasify the material and a long melt pool lifetime allows small pores to merge into large round pores [37]. Between both the conduction mode (orange) as the sweet spot for the highest possible relative densities is located. According to Wang et al., the short lifetime of the melt tracks allows only large gas pores to evacuate [37]. Small and round pores are characteristic of this mechanism [37]. If the energy is locally too low, occasional unshaped pores results. The process window for reaching high relative densities gets smaller with an increase in the hatch distance as well as the layer thickness. Further, occasional unshaped pores become more likely at higher layer thickness. This can indicate an increased sensitivity for process parameter fluctuations at higher layer thicknesses.

Correlations between the process parameters with each other and with the relative density are not detectable. A wide spectrum of relative densities can be reached with all four-factor steps. Further, all factor steps of the full factorial experiment design are clearly visible in the correlation plots as columns of data points parallel to the y-axis, see Figure 7a,b. Between the relative density and the ED, a correlation such as an exponential saturation can be detected, see Figure 7c. It should be noted that the majority of the process parameter combinations result in relative densities above 90%. This can be explained by the high thermal conductivity of the aluminum alloy (AlSi10Mg) used and the low melting temperature range. Thus, a high relative density can already be achieved with relatively low introduced ED values. In order to introduce higher porosities, the energy must be greatly reduced, which further increases process instability. This process instability results not only in a higher porosity but also in a higher standard deviation. The fact that a large number of relative densities can be achieved with a single ED may indicate a significant influence of the selected process parameters and their interactions, see Figure 7d. At this point, the ED can be confirmed as a key parameter for a rough estimation of the achievable relative density for a process parameter set [6]. However, further interactions have to be considered and investigated in detail.

### 4.2. Data Analysis

The database contains 388 instances after adding data from previous experiments (null values excluded), the four process parameters as well as the ED and the resulting relative densities. The distribution of each variable becomes apparent in the histograms, see Figure 8. 

Mainly, the right-side hanging distribution of the layer thickness, the laser power, and the ED get into focus, see Figure 8a–e. The relative density exhibits a left-side hanging distribution, see Figure 8e. These formations indicate that the variables with low densities are underrepresented in the database. This can affect the machine learning training negatively, because missing points cannot flow into the regression model. Complementary, the density plot of the relative density is a sign of a wide range of possible process parameter combinations reaching relative densities above 95%. This can become critical for solving the hypothesis of predicting process parameters for target densities primarily in the case of reaching lower density values. Further, the full factorial experiment design conditionals only four parameter steps for each variable. Even with additional data, the cardinality of the four process parameters is in the area of categorical variables, restricting the reachable model quality further. So, the cardinalities are between six (layer thickness) and 44 (scan speed). The volume energy as well as the relative density own cardinalities of 343 and 279. Constrained by the possible parameter space of the used SLM 125 HL invalid outliers with laser powers > 350 W are removed. Further, density values < 70% and hatch distances > 0.25 mm are detected as valid outliers and are removed too. Thus, a database with 350 instances results.

### 4.3. Forward Modeling Problem-Density Prediction

First, a simple regression model for calculating the resulting relative density by the ED according to Figure 7c is created, see Figure 9a. Using this model, a set of artificial equally distributed process parameters and relative densities is built. For the model training, 280 instances and for the test 120 instances are used. By this, an ANN with 4 inputs and 1 output neuron with two hidden layers including 8 and 4 neurons is trained. The neurons of all layers are activated by a sigmoid function. By use of the Adam algorithm [38] and the mean squared error loss [39], a test prediction error of 0.17% (0.12% for training data) and an R^2^-value of 0.98 (for training and test data) can be obtained, see Figure 9b. If using the ReLU activation function for the hidden layers, a much higher error occurs, see Figure 9c.

**Proof** **of** **Hypothesis** **1.**At this point, hypothesis 1 can be confirmed regarding the artificial data. By the use of machine learning algorithms, especially ANNs, the relative density can be predicted by LPBF process parameters. □ 

### 4.4. Backward Modeling Problem-Process Parameter Prediction

The first approach mentioned in hypothesis 2.1 is a database. The trained forward model is used to predict relative density values for an incremental varied process parameter space. Each parameter is increased in 9 and 18 steps from the minimal to the maximal value. Once the database is calculated, process parameters can be selected by the relative density, the layer thickness, and the BR. The predicted process parameters reach the targeted relative densities, see Figure 10c, because all combinations of the database are synthesized by the use of the trained forward model. The prediction error of the forward model influences the quality of both backward models. The database is also able to rebuild the ED distribution regarding the test data, see Figure 10d. In comparison to the ANN model, the EDs are slightly higher, compare Figure 10d,b. The mean BRs are 3% to 30% lower than the ones of the ANN model. Increasing the number of parameter steps also increases the mean BR since more possible solutions can be considered. 

Secondly, for the prediction of process parameters according to hypothesis 2.2, a relative density, as well as a desired layer thickness, are used as inputs. The laser power, scan speed, and hatch distance will be predicted. For this, a model architecture of 2 input neurons, 2 hidden layers with 4 and 8 neurons, and 3 output neurons is chosen. With the use of artificial data, the concatenated model can learn the inverse problem in less than 10 k training cycles. As shown in Figure 10a, the process parameters predicted by the inverse model reach the targeted densities with a prediction error of 0.09%. Additionally, the model is able to depict the relatively simple ED distribution of the artificial test data set, see Figure 10b. The prediction error depends also on the quality of the chosen forward model. In contrast to approaches such as INNs [21] or non-dominated sorting genetic algorithm (NSGA-II) and modified variational autoenconder (MVAE) [18], the concatenated ANN model does not unfold the complete possible result space. For every pair of target density and target layer thickness, only one process parameter prediction exists. The mentioned models, also the shown database approach [13], unfold the whole result space, so constraints are needed to reduce the number of possible solutions. These constraints can be included in the concatenated ANN training reducing the computing time for process parameter prediction. 

For a constant layer thickness of 0.05 mm, a mean BR for relative densities between 70% and 99% of 18.91 mm^3^/s is reached. By adding a hint to the error function of the model, training the build rate can be increased by 12.5% up to 20.45 mm^3^/s. Two types of hints are formulated, a linear (*MLin*) and a squared (*MSQ*): (2)$$MSQ=\frac{BR^{2}-2\cdot BR+1}{n_{batch}}$$
(3)$$MLin=\frac{BR-2\cdot BR+1}{n_{batch}}$$
where *BR* is the build rate calculated from the predicted process parameters and *n_batch_* is the batch size. These terms are weighted by a beta value (<1) and added to the mean squared error of the relative density as presented by Alpaydin [29]. Both types result in an increased mean BR and a slight increase in the prediction error from ~0.11% to ~0.16%. Higher beta values do not increase the build rate further but reach faster an unwanted saturation of the model. The same effect can be observed by using the *MLin* hint.

The differences between both approaches become visible by looking at the predicted process parameters in detail. In the following, process parameters are generated for relative densities from 70% to 99% at a constant layer thickness of 0.05 mm using both models. For both models, the process parameters show an increasing ED at higher relative densities. For the database approach, the noise is higher compared to the ANN, see Figure 11a.

For the process parameters, the database approach shows strong jumps of the process parameters over the relative density, as exemplified by the laser power and the track distance, see Figure 11b,c. The effect can be attributed to single case mapping of the database. These results show that this approach is able to reproduce the origin distribution (ED and relative density), but does not contain information on the concept of process parameter optimization for the present process. As visible in Figure 10c, the process parameters resulting from the database approach reach the correct relative densities as recently shown by Park et al. [13]. Since the resulting densities are predicted by the forward model, there could be a nexus of both models’ errors, shown by the jumps of the resulting process parameters. The ANN model as a mean value learner shows here more continuous results. It selects a higher laser power over the entire range of relative density values and a decreasing hatch distance with increasing relative density. Increasing the number of steps in the database approach from 9 to 18 results in a higher average BR which corresponds to that of the ANN model. In addition, the size of the jumps in the individual process parameters is reduced and the model approaches the inverse ANN.

Even with 18 levels and thus more than 100 k combinations (Park et al. [13] used 73 k combinations), the database approach still shows considerable jumps in the results. The reduction in the jumps can be explained by the fact that more possible process parameter combinations are available, resulting in a mean value approximation. Since not every density is exactly represented in the database, a selection must be made according to the target density with soft limits. This limit must be selected the softer, the fewer data are contained in the database. In addition, the calculation time increases with an increasing number of database instances.

**Proof** **of** **Hypothesis** **2.**The inverse problem of process parameter prediction can be solved using both a database and a concatenated inverse ANN model. Both models show that they can learn and rebuild the initial distribution related to the ED in a problem-specific manner. In addition, the models can be used to maximize the BR over the entire range of relative densities studied compared to the baseline distribution. For an in-process application, the concatenated ANN shows a higher potential due to the smoothed trajectories as well as an increased tractability of the learned strategy. Thus, the ANN model presents a maximization of the laser power and adjusts the required energy densities via the hatch distance and the scan speed. On the other hand, no generally valid strategy can be determined for the database approach. □

### 4.5. Real Data Application

After proving the solvability of the inverse problem using synthetic data, the methods are adapted to the real data collected in the process. The data set presented in Section 4.2. is extended to 700 instances considering the process noise with artificial normally distributed noise. Subsequently, it is split into 490 training and 210 test instances. A maximum noise of 5% is assumed for the process parameters. Density noise is assumed to increase linearly towards lower relative densities. The increase in the standard deviation is calculated according to:(4)$$\sigma_{\rho }=-0.16\cdot \rho +16.16,$$
where ρ represents the original relative density value and $\sigma_{\rho }$ represents the assumed standard deviation for the density value. Using the calculated standard deviations, a random normally distributed noise is applied to the original data. Model training shows that considerably more training cycles are required and that an architecture with 8 and 4 neurons in the inner layers leads to highly noisy results. With increasing model complexity, more neurons in the inner layers, the results for the models improve. This is especially evident for the training data. For the test data, there is also a decreasing scatter, which, however, exceeds that of the training data considerably, see Figure 12. Because of the noise of the real data, the mean absolute error loss is used for model training. This loss function is more insensitive against noisy data [39].

Due to the split into training and test data and the factor step-based experimental design, there may be increased gaps that cannot be covered in the training process. Thus, the high dispersion of the predicted densities for the test data may result from the low variance of the data set. A test series with higher variance should lead to improved results, which should be comparable to those of the purely synthetic data. Further improvements could be achievable with the implementation of process and material models (PDE). This way, the model can benefit from data and knowledge combined in a PINN [24,40] also shown for melt pool fluid dynamic prediction [41].

For the inverse model, the forward model from Figure 12c is used. With real data and more neurons, the training times are considerably higher. The result with 40 and 60 neurons on the two inner layers is shown in Figure 13. This setup is chosen because it represents the inverse setup of the forward model. If a smaller number of neurons is chosen for the backward model, the results deteriorate, sometimes significantly. As shown in Figure 13a, the process parameters predicted by the backward model, according to the forward model, reach the specified relative densities. The error influence of the forward model has to be taken into account. The resulting ED of the predicted process parameters is oriented to the minimum of the range given by the experimental data, see Figure 13b.

If the relative densities from the test data set are applied to the model, as well as the associated layer thickness, the predicted process parameters according to the forward model meet the specifications with a very high accuracy. If the entire considered density range between 70% and 99.5% at a constant layer thickness of 50 µm is used for the prediction of process parameters, a strong model deviation for relative densities below 75% is shown, see Figure 14a. The laser power is chosen almost constantly by the model, slightly decreasing towards higher densities. The hatch distance is lowered considerably at higher relative densities and thus increasing ED is implemented, see Figure 14b,c.

The accuracy of the backward model depends on the accuracy of the forward model. The backward model can never be better than the forward model for the prediction of the relative density. Thus, with the existing experimental data, the model cannot be meaningfully improved and generalized. More data points are needed at this point. The presented approach can be implemented in practice if the parameter space is constrained. It is assumed, but still not proven, that a process parameter set prediction model is machine and material specific. Therefore, a parameter predictor must be derived from previous experimental data collected and processed as described in this paper. It is worth noticing that the presented model is developed on only small specimens. For future works, the influence of part geometry and volume needs to be taken into account. The residual stresses could further be an additional constraint for the inverse model. Next to increasing the build rate, this can improve the process stability decreasing the amount of scrap parts. 

### 4.6. Summary of the Results

Boundaries of the process window reached with the statistical test series and different kinds of pores and mechanisms could be mapped.Problems with statistical test series for machine learning are detected and evaluated for the article’s target of linking the relative density and the LPBF process parameters.Theoretical solvability of the inverse problem evaluated by synthetic data for both model approaches (concatenated ANN and database).Database approach shows strong jumps for chosen process parameters, while the concatenated ANNs choosing process parameters smooth and strategicBy adding hints, the concatenated ANN model is guided by learning to increase the build rates of predicted process parameters.Concatenated ANNs could learn real data problems, but pure quality and fuzziness of the real data worsen the results of the models.

## 5. Conclusions

It can be stated that both investigated model approaches can be used for solving the inverse problem considered; the prediction of process parameters for given relative densities in the LPBF process. The novelty of this work is the concatenated ANN architecture, which is also trained with real data and provides a technological benefit through process optimization. Despite learning the forward model of a synthetic correlation and although the predicted process parameters follow the ED density correlation model, from an expert point of view, the database approach shows arbitrary jumps in process parameter selection. The ability to reach the target density can be doubted. For the synthetic data, the concatenated ANN model is able to predict the relative density with an R^2^ value of 0.98. The backward model is trimmed to higher build rates by the formulation of a hint during the training process. By this, a 12% increase in the mean BR is possible.

The models with real data, which have significantly lower variance, show significantly worse results. Despite artificial noise, the gaps between the data points are too large for a generalized and robust model. Thus, it can be stated that a statistical experimental design is suitable for a process investigation but has too low a data quality for the training of machine learning models. Interpolation between factor levels would be possible, but this results in an almost entirely synthetic dataset similar to the purely synthetic dataset used for model building. Additionally, the results of the models trained with real data are worsened by the noise of the data. While the ED can generally be used to estimate the relative density, a single ED can result in multiple relative density values and higher EDs do not necessarily affect relative densities. Thus, the ED cannot serve as the single indicator for the resulting relative density. Instead, the process–property relationship needs to be understood.

The lack of process parameter predictor models derived from analytical models or simulations enforces a data-driven method. Even physical model-driven trained predictor functions (such as PINNs) rely on specific material and process models, which are not available or accurate enough for an additive metal powder process. Solving the inverse problem of a data-driven model is difficult due to the lack of explainability and tractability of any data-driven model approximation. Most forward models cannot be inverted with any mathematical method. Combining a forward and backward model in a hybrid model that is trained in conjunction can overcome this limitation.

The concatenation of forward and inverse models implies a dependence on the generalization and prediction accuracy of the models. This study shows the theoretical solvability of the inverse problem via concatenated ANNs, but real-world data with higher variance are required to verify the solution. Compared to GANs, the training of the chained models is supervised and related to one feature (here the relative density). By means of a hint, the BR can be increased additionally. Further, there is no need to unfold the whole result space to select a single set of process parameters. Thus, in application, LPBF machine costs for components may be reduced by using a suitable relative density. However, when defining a relative density for specific LPBF components, it is necessary to understand and exploit the interactions between the results of the LPBF process and subsequent process steps, such as heat treatments or hot isostatic pressing, to achieve the desired advantages.

## Figures and Tables

**Figure 1 materials-15-07090-f001:**
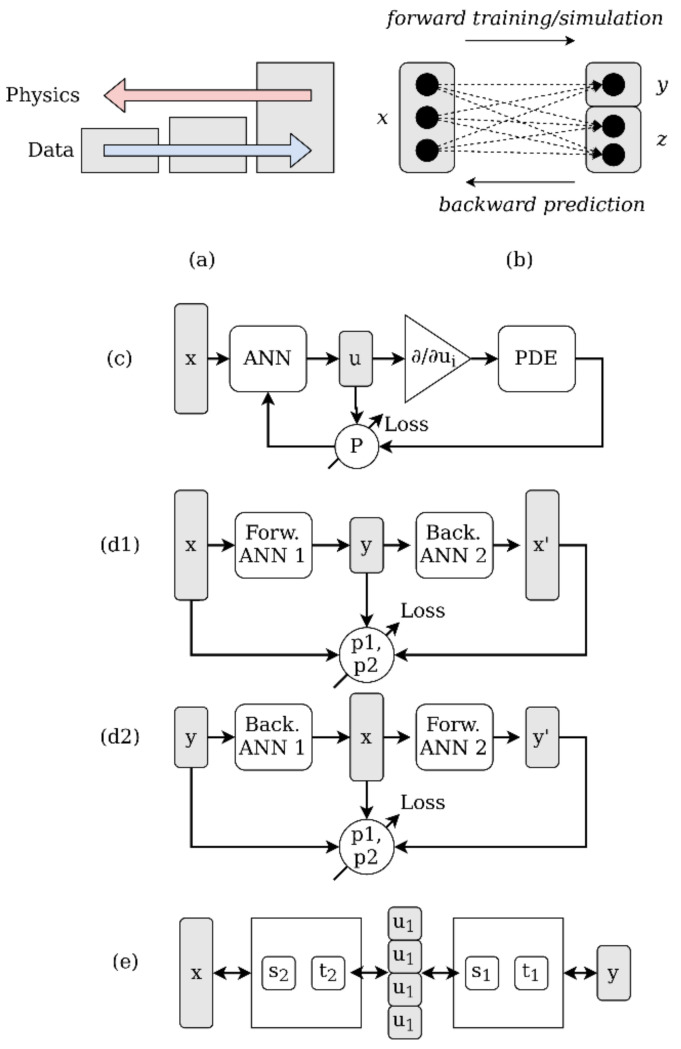
(**a**) Decreasing correlation of physics with increasing volume of data; (**b**) solving the ambiguous inversion problem by adding latent variable z; (**c**) basic concept of physics-informed ANN training; (**d1,2**) forward and backward model chain with closed-loop training; (**e**) affine coupling layer embedding four ANN *s* and *t*.

**Figure 2 materials-15-07090-f002:**
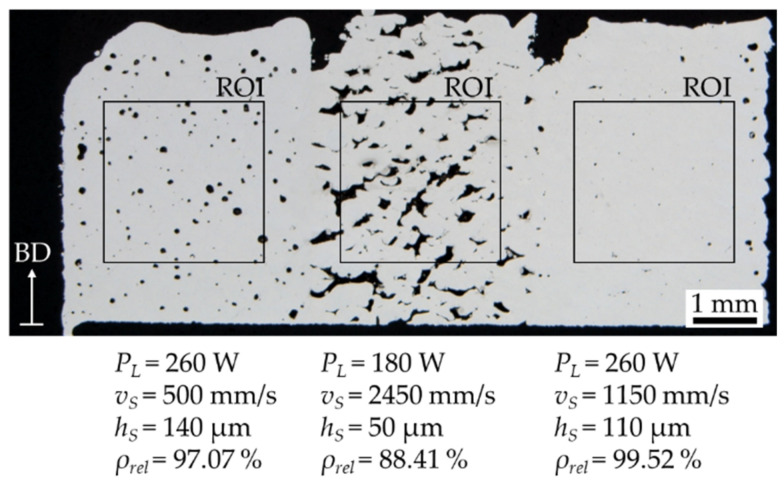
Metallurgical micrograph of three different parameter sets and densities combined in one specimen geometry with a layer thickness of 80 µm and marked region of interest (ROI), as well as the build direction (BD) and the used process parameters.

**Figure 3 materials-15-07090-f003:**
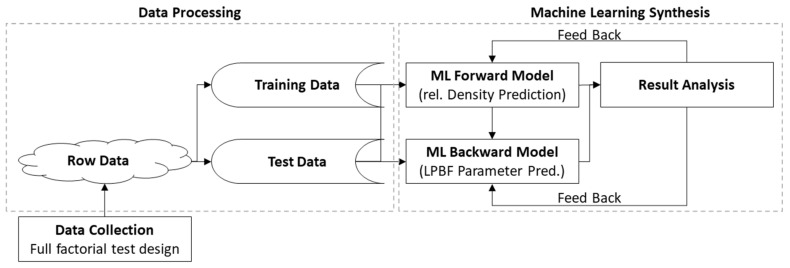
Data flow diagram for the training of the forward (relative density prediction) and the backward model (LPBF process parameter prediction) used in this work.

**Figure 4 materials-15-07090-f004:**
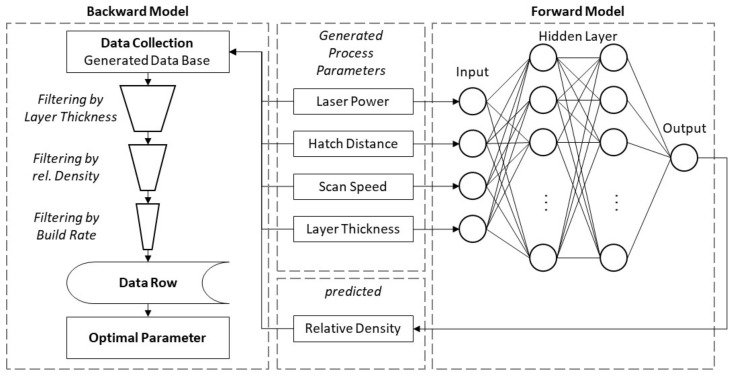
Visualization of the data base approach for solving the inverse modeling problem of predicting LPBF process parameters by the use of a trained forward model according to Park et al. [13].

**Figure 5 materials-15-07090-f005:**
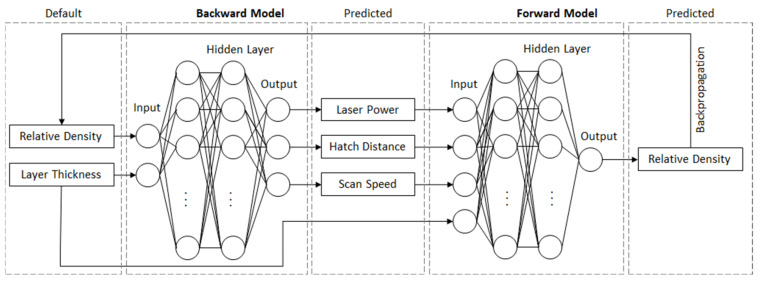
Visualization of the concatenated ANN models for the forward modeling problem (rel. density prediction) and the inverse modeling problem (LPBF process parameter prediction) with in- and outputs used for training and inference.

**Figure 6 materials-15-07090-f006:**
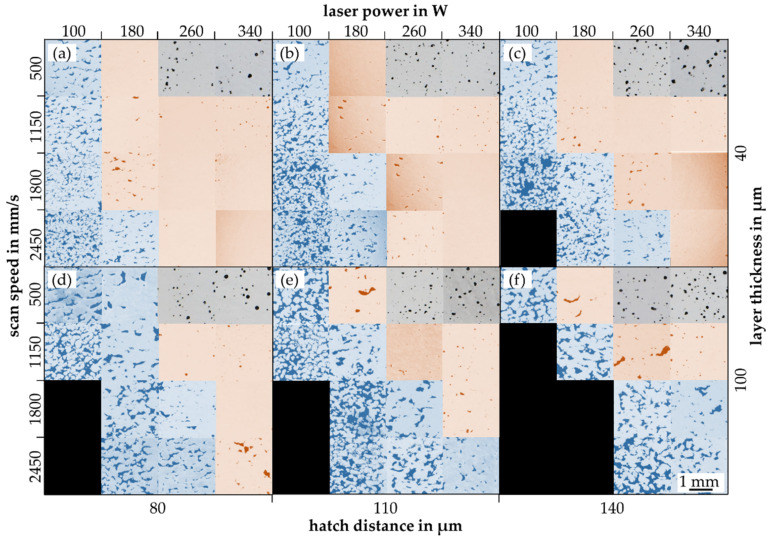
Mapping of metallurgical micrographs for different process parameters colored depending on the pore building mechanism (blue: lack of fusion, orange: conduction, grey: keyhole): micrographs by the laser power and scan speed for a hatch distance and laser power of (**a**) 80 µm and 40 µm; (**b**) 110 µm and 40 µm; (**c**) 140 µm and 40 µm; (**d**) 80 µm and 100 µm; (**e**) 110 µm and 100 µm; (**f**) 140 µm and 100 µm.

**Figure 7 materials-15-07090-f007:**
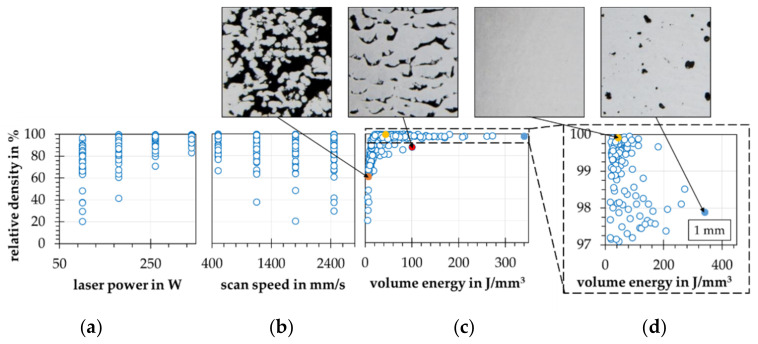
Correlation plots for the relative density depending on selected process parameters resulting from the process parameter examination (**a**) by the laser power; (**b**) by the scan speed; (**c**) by the volume energy (ED); (**d**) by the volume energy (ED) in detail.

**Figure 8 materials-15-07090-f008:**
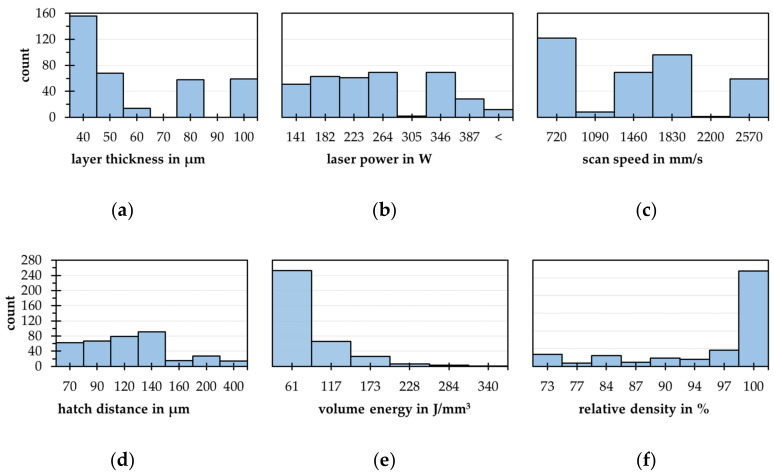
Histograms of all process parameters and the relative densities of the gross data base with 388 Instances (**a**) layer thickness; (**b**) laser power; (**c**) scan speed; (**d**) hatch distance; (**e**) volume energy density; (**f**) relative density.

**Figure 9 materials-15-07090-f009:**
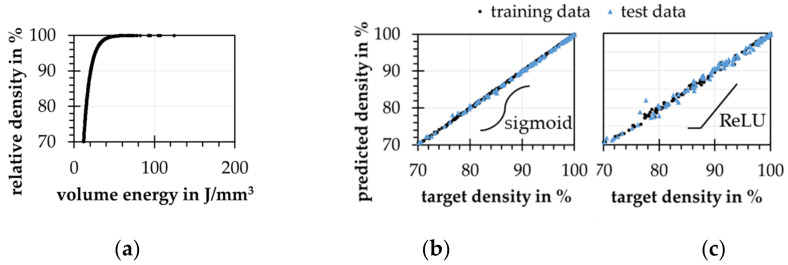
(**a**) artificial relative density over the volume energy density (ED) resulting from the regression model; predicted density over target density of the trained forward model with use of the artificial data (**b**) with sigmoid; (**c**) and ReLU activation.

**Figure 10 materials-15-07090-f010:**
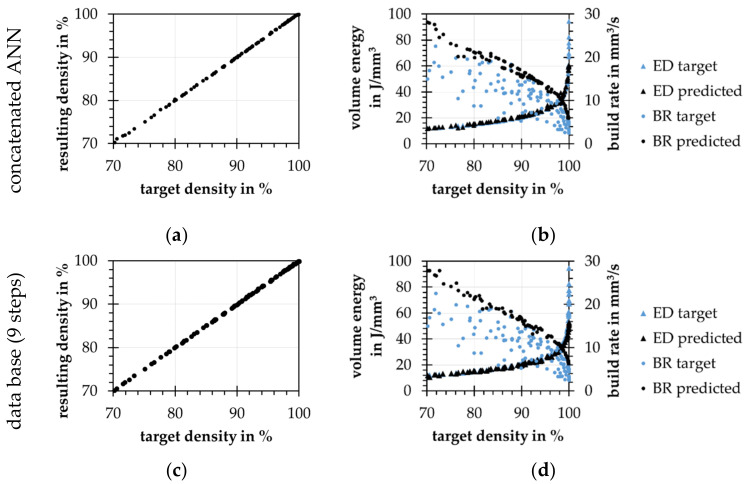
(**a**) resulting densities from predicted process parameters predicted by the trained forward model; (**b**) original and predicted volume energy (ED) and the build rate (BR) distribution over the target density; (**c**) resulting densities over target densities; (**d**) original and predicted ED and BR distributions over the target density.

**Figure 11 materials-15-07090-f011:**
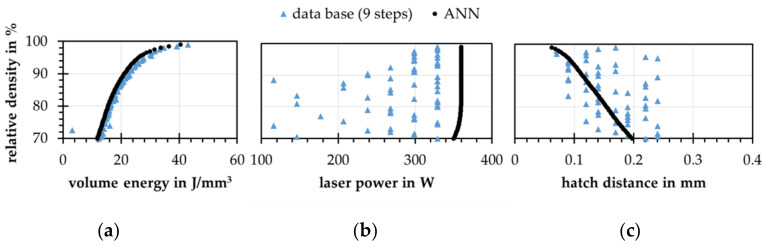
Results of the process parameter prediction for a constant layer thickness of 0.05 mm for both inverse models trained with artificial data, target relative density over (**a**) volume energy; (**b**) laser power; (**c**) hatch distance.

**Figure 12 materials-15-07090-f012:**
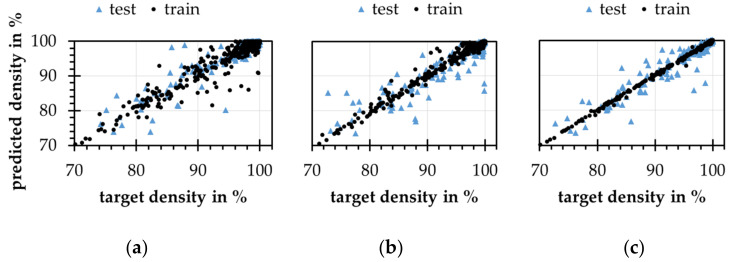
Results of ANN models with the extended experimental data with (**a**) 8 and 4; (**b**) 24 and 12; (**c**) 60 and 40 neurons in the two inner layers.

**Figure 13 materials-15-07090-f013:**
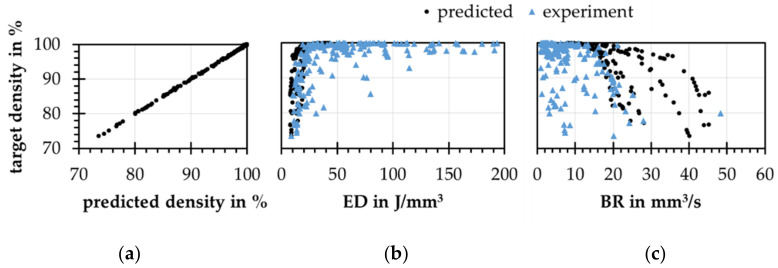
Results of the chained ANN approach with real data, target densities over (**a**) relative densities predicted by the forward model from the process parameters predicted by the backward model; (**b**) ED from the predicted and experimental process parameters; (**c**) build rates of the predicted and experimental process parameters.

**Figure 14 materials-15-07090-f014:**
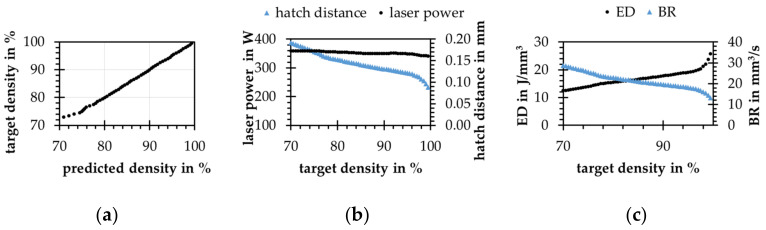
Results of the concatenated ANN approach with the experimental data for given relative densities in 0.5% steps between 70% and 99.5% at a constant layer thickness of 50 µm; (**a**) relative densities predicted by the forward model over the given; (**b**) laser power and hatch distance over the given relative density at a constant scan speed of 3000 mm/s; (**c**) energy density and build rate of the predicted process parameters over the given target density.

**Table 1 materials-15-07090-t001:** Factors and steps for the statistical experiment design without heated build plate.

Factor	Step I	Step II	Step III	Step IV
*D_S_* in µm	40	50	80	100
*P_L_* in W	100	180	260	340
*v_S_* in mm/s	500	1150	1800	2450
*h_S_* in mm	0.05	0.08	0.11	0.14

## Data Availability

Models and data available at: https://doi.org/10.5281/zenodo.7156655.

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
