# Peer review of "Programmable Density of Laser Additive Manufactured Parts by Considering an Inverse Problem"

_materials, 2022, doi:10.3390/ma15207090_

Round 1

Reviewer 1 Report

The paper written in a very good manner and useful to the society. However, I think that there are the following issues to be addressed:

1- List the sections of the paper at the end of the introduction.

2- The literature review is not enough. As the research is related to machine learning, the authors are expected to review more newly published studies, such as 'SVMs classification based two-side cross domain collaborative filtering by inferring intrinsic user and item features', 'a cross-domain collaborative filtering algorithm with expanding user and item features via the latent factor space of auxiliary domains', and 'Training SVMs on a bound vectors set based on Fisher projection' to mention a few. The review of this paper will be enriched if these studies are included.

3- Please justify why your model-approach is feasible?

4- As there are many parameters in the experiments, the authors should explain how to set them properly.

5- Conclusions should be more forceful and elaborated.

Author Response

Manuscript ID: materials-1914767

Title: Programmable Density of Laser Additive Manufactured Parts by Considering an Inverse Problem

Authors: Mika León Altmann, Stefan Bosse, Christian Werner, Rainer Fechte-Heinen, Anastasiya Toenjes

Dear Reviewer,

thank you very much for taking the time to review the manuscript and providing helpful and constructive comments. Please find our detailed response below.

Kind regards

Mika Altmann

The paper written in a very good manner and useful to the society. However, I think that there are the following issues to be addressed:

Thank you for the positive feedback and we are pleased that you mentioned that the topic and our work is useful to the society.

List the sections of the paper at the end of the introduction.

As another author mentioned to give a brief summary of the paper in the end of the introduction, we tried to combine both reviews.

The literature review is not enough. As the research is related to machine learning, the authors are expected to review more newly published studies, such as 'SVMs classification based two-side cross domain collaborative filtering by inferring intrinsic user and item features', 'a cross-domain collaborative filtering algorithm with expanding user and item features via the latent factor space of auxiliary domains', and 'Training SVMs on a bound vectors set based on Fisher projection' to mention a few. The review of this paper will be enriched if these studies are included.

The papers you mentioned enriched the paper exceedingly. They offer a different approach and illuminate the topic from a different point of view. Both were added.

Please justify why your model-approach is feasible?

We tried to clarify how the approach is build up and why it works. I hope that our additional remarks are satisfactory.

As there are many parameters in the experiments, the authors should explain how to set them properly.

We described more precisely how the parameters are set and why there is no single proper way for the process parameter selection. If only the relative density as target property is considered, there are multiple options of process parameters reaching identical results. Only with constraints or additional target properties, a proper selection can take place.

Conclusions should be more forceful and elaborated.

The conclusions are revised and elaborated now. We tried to find a good middle ground between the comments of all reviews. Thank you for your input, the manuscript is clearly improved now.

Reviewer 2 Report

The paper presents an approach to predict the achievable relative density from given process parameters in L-PBF process. Also, models are proposed to solve the inverse parameter prediction problem for a target density.

Please note the below questions/ recommendations:

1. How would the geometry/ part thickness variations affect these models? In other words, would this model be applicable to or remain unchanged for any part geometry? 

2. Abstract: Please precisely describe what practical benefits this extensive study/ approach would bring given different part geometry, materials, desired output parameters such as dimensional stability, mechanical strength, surface quality from the L-PBF process. How would the approach presented in this paper be implemented practically?

3. Materials and Methods: (line 209-210): "Due to age-related inefficiency, the laser power for this examination is limited to 350 W". Please elaborate.

4. References; Reference #9 has not been cited. References citations are not in the sequence they appear in the reference list.

Author Response

Manuscript ID: materials-1914767

Title: Programmable Density of Laser Additive Manufactured Parts by Considering an Inverse Problem

Authors: Mika León Altmann, Stefan Bosse, Christian Werner, Rainer Fechte-Heinen, Anastasiya Toenjes

Dear Reviewer,

thank you very much for taking the time to review the manuscript and providing helpful and constructive comments as well as interested questions. Please find our detailed response below.

Kind regards

Mika Altmann

The paper presents an approach to predict the achievable relative density from given process parameters in L-PBF process. Also, models are proposed to solve the inverse parameter prediction problem for a target density.

Please note the below questions/ recommendations:

  1. How would the geometry/ part thickness variations affect these models? In other words, would this model be applicable to or remain unchanged for any part geometry? 

Thank you for your question. We added our assessment on this topic. Since the model is only trained on small specimens the model cannot be used for parts with a strongly deviating volume with same accuracy. The adaption of volumetric and geometric influences are on our agenda for future work. The physical connections are the same for different volumes admittedly with different cooling rates depending by the part volume. Thereby training a model with more features (process parameters, geometry and volume properties) can be trained improving the usability for practical implementation.

  1. Abstract: Please precisely describe what practical benefits this extensive study/ approach would bring given different part geometry, materials, desired output parameters such as dimensional stability, mechanical strength, surface quality from the L-PBF process. How would the approach presented in this paper be implemented practically?

The point you adressed is very important. We have added our estimation. With some more studies machine learning model can get a proper method improving the additive manufacturing process in many fields of material and part properties.

  1. Materials and Methods: (line 209-210): "Due to age-related inefficiency, the laser power for this examination is limited to 350 W". Please elaborate.

The sentence was misleading. 350 W is the limit we set for the experiments, we clarified the passage.

  1. References; Reference #9 has not been cited. References citations are not in the sequence they appear in the reference list.

The reference content was already listed in the test. Due to formatting the reference number was wrong, it is corrected now.

Reviewer 3 Report

The presented topic “Programmable Density of Laser Additive Manufactured Parts 2 by Considering an Inverse Problem” concerns an important area of additive manufacturing. However, it required revision before publication in following areas:  

  • Abstract needs to be revised. Add both qualitative and quantitative findings in abstract
  • Keywords: Elaborate LPBF and remove process parameter prediction, density prediction.
  • In introduction, it is necessary to give brief introduction about AM and its types. Following articles will be useful for the same: https://doi.org/10.1016/j.jmrt.2022.08.074 ;  https://doi.org/10.1007/s00170-015-7576-2
  • Need to give more emphasize on research gap of present work by elaborating the recent work carried out by researchers. Article lacks in this area
  • Briefly describe the summary of your work in last paragraph of introduction.
  • How the parameters were selected?
  • Too many symbols were used. Authors should add nomenclature section after the conclusion.
  • Explain the sub-figures for figure 6. It is necessary to add figure captions e.g. 6a, 6b….
  • In results and discussion section, compare your findings with past studies and give reasonable agreement from literature
  • Conclusion section is too large. Remove references from conclusion. Also, not necessary to give reference of any figure/table in conclusion.
  • References are not as per the journal format. Correct it

Author Response

Manuscript ID: materials-1914767

Title: Programmable Density of Laser Additive Manufactured Parts by Considering an Inverse Problem

Authors: Mika León Altmann, Stefan Bosse, Christian Werner, Rainer Fechte-Heinen, Anastasiya Toenjes

Dear Reviewer,

thank you very much for taking the time to review the manuscript and providing helpful and constructive comments. We are pleased you mentioned the importance of the topic. Please find our detailed response below.

Kind regards

Mika Altmann

The presented topic “Programmable Density of Laser Additive Manufactured Parts 2 by Considering an Inverse Problem” concerns an important area of additive manufacturing. However, it required revision before publication in following areas: 

Abstract needs to be revised. Add both qualitative and quantitative findings in abstract

You are absolutely right, we added the findings in the abstract.

Keywords: Elaborate LPBF and remove process parameter prediction, density prediction.

The Keywords are revised and we keywords facing more LPBF.

In introduction, it is necessary to give brief introduction about AM and its types. Following articles will be useful for the same: https://doi.org/10.1016/j.jmrt.2022.08.074 ;  https://doi.org/10.1007/s00170-015-7576-2

Thank you for your Feedback. We added more brief introduction in additive manufacturing and some metal relevant types.

Need to give more emphasize on research gap of present work by elaborating the recent work carried out by researchers. Article lacks in this area

We worked out the research gap more. We added some literature and pointed out the interest in demand oriented process parameter for AM.

Briefly describe the summary of your work in last paragraph of introduction.

We added a briefly summary of the work and it´s chapters to the introduction.

How the parameters were selected?

The process parameters were chosen as wide as possible to reach a maximum range of different relative densities. Also we wanted to reach the process limits. By use of the min and max values, a full factorial design of experiments test design is created and build.

Too many symbols were used. Authors should add nomenclature section after the conclusion.

As there are many symbols for sure, we added an abbreviation section like you mentioned.

Explain the sub-figures for figure 6. It is necessary to add figure captions e.g. 6a, 6b….

Thank you for the feedback to the figure. To clarify the figure we revised the caption and numbered the sub-figures.

In results and discussion section, compare your findings with past studies and give reasonable agreement from literature.

For improving the results and discussion section, we added more literature and references as you mentioned.

Conclusion section is too large. Remove references from conclusion. Also, not necessary to give reference of any figure/table in conclusion.

We removed the references form conclusion section. As other reviews required more detailed conclusions we tried finding the best way for both reviews. Hopefully you are fine with it.

References are not as per the journal format. Correct it

The Reference Section is corrected now.

Round 2

Reviewer 3 Report

All the changes of the previous round has been addressed by the authors. In my opinion, the article can be accepted in the present form.